# Towards Ontology-Driven Multi-Hop Benchmark for Corporate GraphRAG Systems

Ivan E. Kirpichev[1,*,†], Eldar R. Kurmanaliev[1,*,†], Fedor V. Bushmelev[2], Maxim V. Abramov[2], Anastasiia A. Korepanova[2], Nikolay O. Nikitin[1] and Anna V. Kalyuzhnaya[1]

*¹ITMO University, Kronverksky Pr. 49, Saint-Petersburg, 197101, Russian Federation*

*²SPC RAS, Saint-Petersburg, 199178, Russian Federation*

### Abstract

Existing GraphRAG benchmarks suffer from two evaluation problems: static corpora already memorized by modern LLMs, and synthetically generated questions whose answers may not be grounded in the data. We present an automated framework that constructs a dynamic, news-enriched corporate knowledge graph and generates benchmark questions whose ground truth is directly validated against the database. The graph combines a strict ontology of S&P 500 companies with executives, funds, products, resources, geographic entities, and a recent news stream. Questions are produced by extracting real paths from local subgraphs and using an LLM only to translate verified queries into natural language. The resulting dataset of 4,998 question–answer pairs across six complexity levels is used to compare Vanilla RAG, LightRAG, MS GraphRAG, and HippoRAG2.

### Keywords

GraphRAG, Multi-Hop Question Answering, Knowledge Graph, Ontology-driven RAG, Financial-domain Knowledge Graph

## 1. Introduction

Retrieval-Augmented Generation (RAG) systems have become the dominant paradigm for integrating external knowledge into large language models (LLMs), reducing the risk of hallucinations and ensuring the relevance and timeliness of generated responses. Recently, researchers' attention has shifted toward graph-based systems (GraphRAG) [1], which employ structured knowledge graphs to model hierarchical relationships between concepts. This approach is assumed to enable more coherent knowledge retrieval and more effective structural reasoning compared with standard vector-based methods.

Nevertheless, the widespread adoption of GraphRAG faces a paradoxical evaluation problem. As shown in recent studies, such as GraphRAG-Bench [2], graph-based systems often underperform classical vanilla RAG systems on real-world tasks.

An additional, yet no less critical, challenge lies in the nature of the data itself. Most public graph benchmarks are static and rely on historical corpora, such as financial reports from 2022–2024. The use of such datasets is associated with a high risk of data contamination or memorization: the tested LLMs may retrieve correct answers from their own training data rather than from the provided RAG context [3]. Moreover, static benchmarks do not reflect the real needs of businesses, especially in the financial domain, where dynamic information flows must be analyzed and where relationships between companies continuously change under the influence of news.

To address these challenges, a tool is required that can generate complex test scenarios based on up-to-date data previously unknown to the model, while guaranteeing the complete concrete resolvability of these scenarios within the graph.

In this work, we present an automated framework designed to address both problems: data obsolescence and factual bias in evaluation. Our system constructs a ground-truth financial knowledge graph

*IJCAI-ECAI 2026 Joint Workshop on GENAIK and NORA, August 15-17, 2026, Bremen, Germany*

*Project repository: https://github.com/Elik-sir/Ontology-Driven-Multi-Hop-Benchmark-for-Corporate-GraphRAG-Systems-*

*Corresponding author.

†These authors contributed equally.

✉ 507011@niuitmo.ru (I. E. Kirpichev); 506810@niuitmo.ru (E. R. Kurmanaliev)

🌐 https://github.com/KirpichevIvan (I. E. Kirpichev); https://github.com/Elik-sir (E. R. Kurmanaliev)

by enriching a static ontology with a stream of up-to-date, cleaned news data, and then automatically transforms this topology into a set of validated test data. A key feature of the proposed question generator is the transition from text-based generation to graph path-based generation, with concrete validation of each scenario through the execution of Cypher queries against the database management system.

The main contributions of this work are as follows:

- **Dynamic ground truth:** We propose a pipeline for the automatic construction of a ground-truth knowledge graph that combines a strict hierarchy based on the GICS standard with a continuous stream of news data.
- **Multi-level question generation:** We develop a generator architecture capable of producing questions across five levels of complexity, ranging from the extraction of simple facts to analytics.
- **Concrete verification:** We develop a mechanism for rigorous validation of ground-truth answers through direct interaction with the database management system via Cypher-based validation. This completely eliminates the risk of LLM hallucinations at the benchmark construction stage.
- **Objective evaluation of GraphRAG:** We demonstrate the practical utility of the proposed benchmark by comparing modern RAG systems. The use of dynamic data makes it possible to identify the structural advantages and current limitations of graph-based approaches under conditions in which the data are unavailable to models through pretraining.

## 2. Related Work

### 2.1. Financial QA and the Problem of Data Contamination

Financial data analysis is a challenging task for standard RAG systems, which often make errors when processing dynamic information streams. As shown in the CRAG benchmark [4], even advanced RAG systems exhibit reduced accuracy on financial and rapidly changing data. Attempts to create specialized datasets, such as FINANCE BENCH [5], FinBen [6], and FINDER [7], emphasize that real-world analyst queries are highly complex and require deep synthesis; however, these datasets do not provide graph connectivity.

Recent advances have demonstrated the effectiveness of knowledge graphs in this domain. The authors of FinReflectKG [8] proposed an automated pipeline for extracting KGs from corporate reports of S&P 100 companies and testing multi-hop queries. However, this approach relies on static reports from 2022–2024. The use of such historical corpora is susceptible to the risk of data contamination or memorization: LLMs often retrieve answers from their own training data rather than from the provided context. Dynamic-KGQA [9] highlights the need for adaptive datasets. A partial solution is proposed in DRAGOn [10], where a periodically updated textual corpus, namely news data, is used. Our work extends this idea by combining a dynamic news stream with strict domain specificity and ontology.

### 2.2. GraphRAG and Ontological Integrity

Recently, there has been rapid progress in state-of-the-art GraphRAG solutions, such as Microsoft GraphRAG [1], LightRAG [11], and HippoRAG [12]. Traditional baseline approaches in these systems are often based on community detection algorithms for bottom-up clustering of nodes. As noted in OntoRAG [13] and Text2KGBench [14], this approach disrupts the ontological integrity of the original data. The construction of formal hierarchical structures is necessary for global understanding and multi-step reasoning.

Existing corporate graphs, such as CompanyKG [15], also provide large-scale structures of intercorporate relationships, but they are often anonymized and static.

## 2.3. Datasets for Multi-hop Question Answering Benchmarks

To evaluate systems' ability to perform complex retrieval, datasets such as HotpotQA, 2WikiMultiHopQA, and MuSiQue are commonly used [16, 17, 18]. In addition, more recent benchmarks for evaluating complex pipelines have been proposed, including MultiHop-RAG and UltraDomain [19, 20].

**Limitations:** Existing multi-hop benchmarks have several critical shortcomings when applied to GraphRAG. First, questions in benchmarks such as 2WikiMultiHopQA and MuSiQue are often constructed artificially on the basis of rigid rules and logic, which distances them from natural queries in real-world scenarios. Second, these benchmarks predominantly contain factual questions with short answer formats, such as names and dates, and therefore exhibit highly limited complexity. Third, modern datasets such as MultiHop-RAG and UltraDomain place excessive emphasis on retrieval difficulty, that is, finding scattered facts, while neglecting the complexity of the reasoning process itself, namely the need to synthesize interconnected facts into contextually grounded solutions.

## 2.4. Automated Data Generation Using LLMs

With the emergence of powerful large language models, a trend has developed toward the automatic generation of synthetic datasets for QA, for example through Evol-Instruct methods [21]. In the context of graphs, researchers propose using LLMs to generate questions directly from an ontology.

**Limitations:** Approaches that generate questions solely on the basis of the graph schema are highly prone to hallucinations. An LLM may formulate a multi-hop question that is logically valid but has no concrete solution because factual relationships between specific nodes are absent in the real data. The "gold standard" or ground truth generated under such conditions often turns out to be merely a plausible model-generated guess rather than a verified fact.

## 2.5. Positioning of the Proposed Benchmark

The literature review reveals a lack of tools capable of automatically generating datasets for private knowledge graphs while guaranteeing the concrete executability of queries and covering different levels of reasoning complexity, from simple facts to deep analytics.

The architecture proposed in this work overcomes the limitations described above. Through the stage of grounding on paths and validation by executing Cypher queries, the system guarantees that the generated multi-hop scenarios have precise concrete confirmation in the data. This completely eliminates the risk of LLM hallucinations at the stage of ground-truth construction and enables an objective measurement of the real capabilities of GraphRAG algorithms in complex aggregation tasks.

# 3. Proposed Benchmark

## 3.1. Graph Construction

The automatic generation of validated questions and the evaluation of RAG systems require an absolutely reliable source of truth. In this study, we developed a pipeline for the automated construction of a financial knowledge graph, whose architecture consists of creating a static ontological core followed by event integration. The overall architecture of the proposed pipeline is illustrated in Figure 1. The resulting heterogeneous graph constructed using our approach contains 49,017 nodes and 70,162 relationships.

### 3.1.1. Formation of the Ontological Core (Static Ontology Core)

To avoid the data fragmentation problems characteristic of text clustering methods, the foundation of our graph is constructed in a top-down manner based on a strict hierarchy. The final ontology schema of the graph is presented in Figure 2. The foundation is the list of companies included in the S&P 500

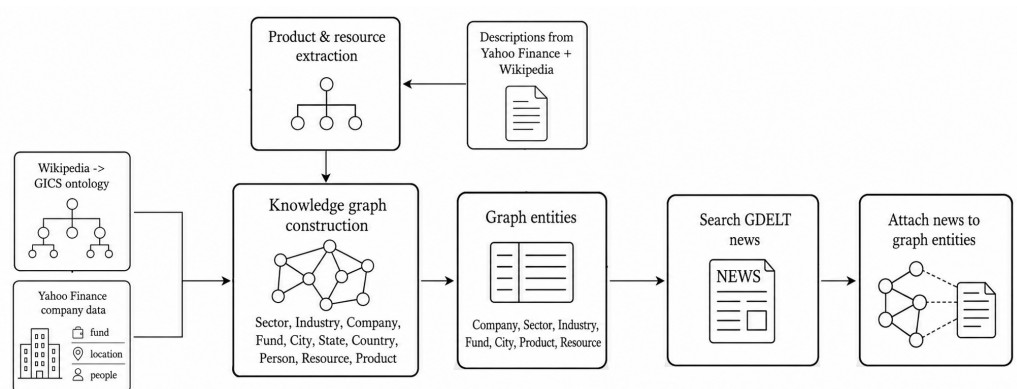

**Figure 1:** Overview of the automated pipeline for financial knowledge graph construction. The process integrates static ontological data and financial metadata with LLM-extracted production topology and a dynamic event stream.

index, structured according to the GICS classification standard, which forms a rigid baseline topology: Sector → Industry → Company.

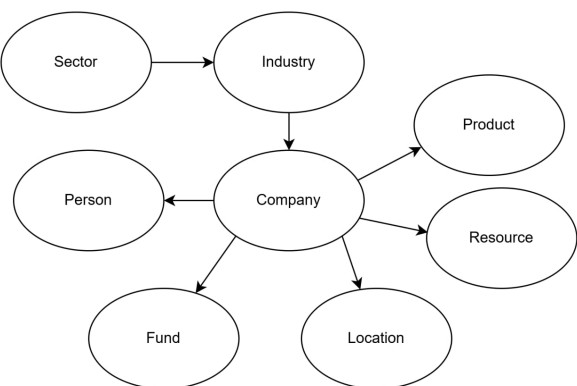

**Figure 2:** Ontology schema of the financial knowledge graph, displaying the GICS-based hierarchy, multidimensional business context (Product, Resource, Person, Fund), spatial entities, and the integration of dynamic News nodes.

To enable the graph topology to generate complex multi-hop analytical queries, the static hierarchy is enriched with a multidimensional business context. Graph expansion proceeds along three key axes:

- **Topology of production relationships:** To formalize operational activity, the LLM *Qwen2.5-7B-Instruct* was employed. This open-weight model was chosen for its state-of-the-art extraction performance relative to its size, cost-efficiency, and the capability for local deployment to ensure data privacy. During the processing of corporate descriptions (Yahoo Finance, Wikipedia), the pipeline analyzed approximately 800,000 tokens and automatically extracted 6,330 *Product* nodes (manufactured goods/services) and 2,774 *Resource* nodes (consumed raw materials/technologies). These nodes were integrated into the database through 9,869 typed relationships (PRODUCES and REQUIRES).
- **Corporate structure:** Based on structured financial metadata, 4,944 key executives (*Person*) and 125 institutional investors (*Fund*) were added to the graph, making it possible to model intersections of interests and assess dependencies through common shareholders.
- **Spatial topology:** The extraction of geographic entities created a layer of 281 locations (*City, State, Country*) for geopolitical analysis.

The presence of such diverse typed relationships creates an ideal heterogeneous environment for

testing the ability of LLMs to perform complex logical reasoning, ranging from the identification of spatial dependencies to the discovery of hidden corporate intersections.

### 3.1.2. Dynamic Event Injection

The main limitation of many existing graph benchmarks is their static nature: they are based on historical document corpora, such as annual reports, whose data may already be compromised by their presence in the training sets of modern LLMs. To address this problem, our pipeline integrates a specialized module that aggregates an up-to-date news stream through the GDELT database [22].

The retrieved events, strictly limited to a chronological window covering the last three months to ensure data uniqueness and novelty, are integrated into the graph through typed relationships, such as NEWS_ABOUT_COMPANY or NEWS_ABOUT_RESOURCE. During data collection, the pipeline successfully injected 33,921 up-to-date news nodes (*News*).

Such temporal enrichment produces a dual effect. First, static entities receive up-to-date situational context, enriching their local semantic neighborhood with event-driven data. Second, news nodes function as "dynamic bridges" by automatically interlinking disparate entities (e.g., a specific company and a raw material) that are simultaneously mentioned within the same news event. This connection is established through the automated creation of multiple incident edges from a single *News* node to all relevant graph entities identified in the text. In our graph, these bridges generated more than 51,000 relationships. As a result, a dense yet logically verified network is formed, opening up opportunities for generating complex temporal and cross-domain benchmark questions.

## 3.2. Benchmark with Question Generation

This section describes the architecture and implementation of the proposed automatic benchmark generator for GraphRAG systems. The system is a pipeline designed to extract the topological structure of a private knowledge graph and transform it into a representative set of validated question–answer pairs.

### 3.2.1. System Overview

The architecture of the proposed benchmark generation framework is illustrated in Figure 3. The pipeline consists of three main stages:

1. **Schema-aware extraction and stratified sampling:** Identifying valid starting points (anchors) and structural paths across the graph.
2. **Hybrid question generation:** A combination of reliability-first deterministic templates for structural queries and LLM-driven generation for complex analytics.
3. **Concrete validation and dataset refinement:** Cypher-based execution and strict ground truth formation.

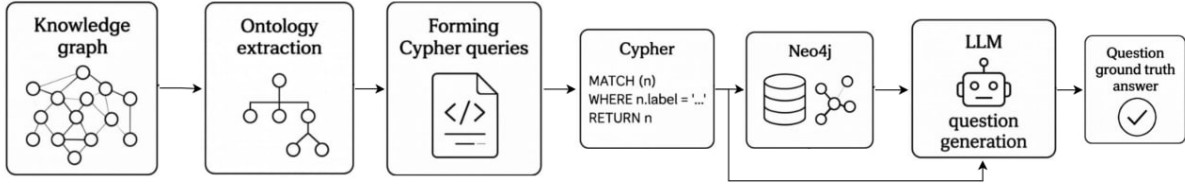

**Figure 3:** Workflow of the automated benchmark generation pipeline, illustrating the transition from raw graph topology to validated question–answer pairs through Cypher-based concrete verification and LLM translation.

### 3.2.2. Schema Extraction and Data Grounding

Unlike traditional benchmark construction methods that rely on unstructured text fragments, our approach is rooted in the formal ontology of the database and begins with a stratified anchor sampling algorithm. Instead of purely random entity selection, the system performs balanced sampling across various node labels and relationship types, preventing data skewness toward overrepresented entities (e.g., highly covered companies) and ensuring diverse topological coverage.

From each selected anchor node, a local subgraph extending 3–4 hops is extracted to facilitate path-based grounding. Within these subgraphs, specific paths—represented as sequences of node labels and relationship types—are identified. Because the theoretical existence of a path in a schema does not always guarantee its presence in concrete data, extracting these actually existing trajectories from the local subgraph is critically important to ensure that all generated multi-hop questions have at least one guaranteed concrete solution within the graph.

### 3.2.3. Logic of Multi-level Question Generation

The core of the generation logic utilizes the specific concrete paths previously extracted from the local subgraphs during the grounding stage. These trajectories are systematically translated into executable Cypher queries, ensuring that every generated question–answer pair is grounded in the actual data topology rather than theoretical schema assumptions. This approach allows for the creation of four distinct question categories:

- **Simple fact questions (1-hop):** Focused on direct attributes or single-hop relationships. The questions are derived from the 1-hop paths.
- **Multi-hop questions (2–4 hops):** These require logical traversal of relationship chains. The system takes the 2–4 hop paths extracted from the subgraph, converts them into complex Cypher patterns, and uses the resulting structure to form the natural language question.
- **Aggregation:** These questions are generated by applying global functions (`count`, `min/max`, `sum`) to the translated Cypher paths, testing the system's numerical reasoning capabilities over the graph.
- **Analytics:** For this type, the entire extracted subgraph context and its corresponding paths are provided to an LLM. While Cypher is used for validation, the final complex question is synthesized to evaluate high-level analytical reasoning.

To eliminate LLM hallucinations and ensure the integrity of the benchmark, every generated case undergoes a rigorous verification and finalization process. First, through Cypher execution validation, each generated query is directly executed in a Neo4j instance, ensuring that only queries returning valid, non-empty, and high-signal results (e.g., excluding null values or trivial counts) are accepted into the final dataset.

Subsequently, the framework employs the LLM exclusively as a translation engine to convert the validated Cypher query into human-readable text. By providing the LLM with the strict structural context of the verified path alongside the resulting Cypher code, the system guarantees that the generated natural language question accurately reflects the underlying data logic without introducing factual errors.

## 4. Experimental studies

This section presents the results of an empirical evaluation of contemporary RAG systems using the proposed dynamic graph benchmark. The experiments are designed to assess the reliability of the generated dataset and to evaluate the effectiveness of graph-based RAG systems across different levels of task complexity.

### 4.1. Experimental Setup

To ensure the reproducibility of the experiments, the evaluation environment was configured as follows. The benchmark generation pipeline produced a final dataset comprising 4,998 question–answer pairs. The dataset is balanced across several levels of complexity and includes 965 simple factual questions requiring one-hop retrieval, 948 two-hop multi-hop questions, 845 three-hop multi-hop questions, 756 four-hop multi-hop questions, 895 aggregation questions, and 589 analytics tasks.

The comparative evaluation considered four retrieval-augmented approaches. The first baseline is Vanilla RAG, a classical vector-retrieval system implemented using Qdrant. The second system is LightRAG [11], a lightweight graph-based RAG framework designed for the rapid integration of local and global contexts. The third system is MS GraphRAG [1], a hierarchical approach that relies on community clustering to support global structural understanding. The fourth system is HippoRAG2 [12], which is based on neurobiologically inspired principles of long-term memory and is intended to process complex relational structures.

In all experiments, the open-weight *Qwen2.5-32B* model was used as the final answer generator. This choice was motivated by the model's strong analytical and reasoning capabilities, which are necessary for interpreting the retrieved subgraphs and producing coherent answers from structured context.

For a comprehensive assessment of RAG-system performance, the evaluation was divided into two independent stages: the retrieval stage and the generation stage. At the retrieval stage, the retrievers were evaluated using *Context Recall*, which measures the completeness with which all nodes and relationships required for answering a question are retrieved, and *Context Relevance*, which assesses the degree to which the extracted subgraph is free from irrelevant informational noise. At the generation stage, the evaluation focused on the ability of *Qwen2.5-32B* to formulate a final answer based on the provided context. The generated answers were assessed using an LLM-as-a-Judge approach based on *Qwen2.5-72B-Instruct*, with the computation of *Faithfulness*, reflecting factual reliability and the absence of hallucinations, and *Answer Relevance*, reflecting the semantic correspondence and completeness of the final response, following the RAGAS framework [23].

### 4.2. Benchmark Quality Assessment

A major limitation of existing approaches to automatic dataset generation using LLMs is the high risk of producing queries that are logically plausible but actually unexecutable, since factual relationships between specific nodes may be absent in the actual data. In this subsection, we evaluate the effectiveness of the proposed validation mechanism, which addresses this problem.

The proposed architecture guarantees concrete resolvability and filtering, thereby fully eliminating the risk of LLM hallucinations at the stage of constructing question–answer pairs. During generation, each formulated query was subjected to strict verification by directly executing the corresponding query in the Neo4j database. Only queries that returned valid, non-empty, and highly informative results, excluding null values, were admitted into the final dataset. As a result, the system guarantees that 100% of the generated multi-hop scenarios have precise concrete confirmation in the data.

To ensure the reliability of natural-language translation and prevent factual distortion in question formulation, the framework uses the LLM exclusively as a translation mechanism for converting a verified Cypher query into human-readable text. By providing the language model with the strict structural context of the verified path together with the resulting Cypher code, the system ensures that the generated natural-language question accurately reflects the underlying database logic without introducing factual errors.

To confirm the linguistic quality of the dataset, a human evaluation was conducted on a random sample of 1,000 generated question–answer pairs. The analysis showed that the strict constraints imposed by Cypher-based verification do not reduce readability: the questions are phrased as natural analytical queries while preserving absolute factual grounding in the extracted subgraphs.

For a comprehensive assessment of system performance, the context retrieval and answer generation phases were evaluated using the proposed benchmark. The combined results, showing the performance

across all task complexities and metrics, are presented in Table 1.

**Table 1**

System Performance Evaluation across task categories. Metrics include Context Relevance and Context Recall (Retrieval phase), alongside Faithfulness and Answer Relevance (Generation phase).

| Task Category | Metric | Vanilla RAG | LightRAG | MS GraphRAG | HippoRAG2 |
|---|---|---|---|---|---|
| Simple | Context Relevance | 0.60 | **0.65** | 0.61 | 0.64 |
| | Context Recall | 0.48 | **0.57** | 0.49 | 0.54 |
| | Faithfulness | 0.40 | **0.65** | 0.61 | 0.63 |
| | Answer Relevance | 0.18 | 0.26 | 0.22 | **0.34** |
| Multi-hop (2) | Context Relevance | 0.44 | 0.54 | 0.57 | **0.61** |
| | Context Recall | 0.45 | **0.56** | 0.51 | 0.54 |
| | Faithfulness | 0.37 | **0.64** | 0.61 | 0.61 |
| | Answer Relevance | 0.05 | 0.25 | 0.22 | **0.32** |
| Multi-hop (3) | Context Relevance | 0.35 | 0.51 | 0.45 | **0.59** |
| | Context Recall | 0.27 | 0.52 | 0.44 | **0.53** |
| | Faithfulness | 0.30 | 0.59 | **0.61** | 0.60 |
| | Answer Relevance | 0.05 | 0.23 | 0.19 | **0.31** |
| Multi-hop (4) | Context Relevance | 0.13 | 0.23 | 0.15 | **0.57** |
| | Context Recall | 0.14 | 0.47 | 0.36 | **0.49** |
| | Faithfulness | 0.30 | 0.53 | 0.51 | **0.58** |
| | Answer Relevance | 0.03 | 0.23 | 0.21 | **0.31** |
| Aggregation | Context Relevance | 0.05 | 0.09 | **0.13** | 0.07 |
| | Context Recall | 0.04 | 0.17 | **0.24** | 0.09 |
| | Faithfulness | 0.14 | 0.34 | 0.29 | **0.37** |
| | Answer Relevance | 0.07 | 0.23 | 0.22 | **0.32** |
| Analytical | Context Relevance | 0.09 | 0.22 | **0.32** | 0.30 |
| | Context Recall | 0.03 | 0.59 | **0.64** | 0.62 |
| | Faithfulness | 0.31 | 0.63 | 0.52 | **0.75** |
| | Answer Relevance | 0.03 | 0.41 | **0.61** | 0.45 |

The comparative analysis showed that failures of classical systems in answer generation, reflected in low *Answer Relevance*, are in most cases caused by the retrieval stage, reflected in low *Context Recall*, rather than by limitations of the language model itself. The introduction of graph structures makes it possible to bridge this gap by providing the generator with a reliable foundation of concretely verified data.

The analysis of the data presented in Table 1 reveals two key trends in the behavior of the studied architectures when processing ontology-oriented queries.

**Expected Underperformance of Vector Search (Vanilla RAG).** At both the context retrieval and the generation stages (Table 1), the baseline Vanilla RAG approach demonstrates the lowest performance across all categories. On complex analytical tasks, *Context Recall* decreases to 0.03, while *Answer Relevance* also drops to 0.03. This result is expected: since the test questions were constructed with strict grounding in the ontology and graph structure of the domain, classical semantic search proved unable to capture multi-level relationships between entities, which led to a critical loss of context.

**Degradation of Graph-Based Systems Due to Ontology Distortion.** Modern graph-based architectures, including LightRAG, MS GraphRAG, and HippoRAG2, demonstrate substantially higher effectiveness compared with Vanilla RAG; for example, *Faithfulness* reaches 0.61–0.65 on simple tasks. However, as the cognitive complexity of the task increases, moving from single-hop queries (Simple) to longer inference chains (Multi-hop 2–4) and aggregation tasks (Aggregation), all three systems exhibit a consistent decline in their metrics.

This degradation can be explained by the algorithmic characteristics of the models themselves. The

systems under consideration do not rely on the reference ontology of the dataset; instead, during document corpus processing, they "break" the original structure and construct their own knowledge graphs according to their internal extraction mechanisms. The resulting topological mismatch between the original ontology, on the basis of which the multi-hop questions were constructed, and the graphs built by the systems leads to a gradual loss of semantic relationships and a decline in answer quality over longer logical distances.

## 5. Conclusions

In this study, a knowledge graph was constructed based on a formal ontology, serving as the foundation for the development of a specialized benchmark. A key feature of the created benchmark is the direct dependence of the question logic on the strict ontological relations and hierarchical dependencies of the subject domain.

The experimental results confirmed the inadequacy of vector RAG systems in tasks requiring the interpretation of the original entity hierarchy: the lack of structural understanding of the data leads to a critical loss of context. Although GraphRAG architectures offer distinct advantages, their performance metrics degrade substantially under high query complexity. This limitation primarily stems from their automated knowledge extraction and graph generation pipelines, which tend to distort or omit the original ontological dependencies. Because this underlying semantic foundation is compromised, current GraphRAG models fail to support the multi-step logical reasoning required in specialized domains with strictly predefined ontologies.

It should be noted that the current iteration of the proposed framework serves primarily as a proof-of-concept. It successfully demonstrates the feasibility of ontology-driven, Cypher-validated dataset generation and exposes the structural vulnerabilities of modern retrieval systems. In a broader context, this approach lays the groundwork for creating more comprehensive and scalable benchmarks, which are essential for the objective evaluation and advancement of corporate GraphRAG solutions.

## Acknowledgments

This research is financially supported by The Russian Scientific Foundation, Agreement № 24-71-10093, https://rscf.ru/en/project/24-71-10093/.

## Declaration on Generative AI

During the preparation of this work, the authors used Gemini in order to: Text Translation, Grammar and spelling check. After using these services, the authors reviewed and edited the content as needed and take full responsibility for the publication's content.

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

# A. Benchmark Data Examples

This appendix provides examples of generated question–answer pairs from the proposed benchmark, demonstrating different levels of structural and analytical complexity. For each case, the natural language question, the corresponding Cypher query used for concrete validation, and the final ground truth answer are presented.

## A.1. Simple Queries

These queries focus on direct attributes or single-hop relationships within the graph topology.

- **Question:** Which industry does Ameriprise Financial, Inc. operate in?
- **Cypher Query:** `MATCH (c:Company {name: "Ameriprise Financial, Inc."})-[:OPERATES_IN_INDUSTRY]->(i:Industry) RETURN i.name LIMIT 10`
- **Ground Truth:** Asset Management & Custody Banks

- **Question:** What company does Aristotle Capital Management, LLC hold a stake in, and what is the ownership percentage?
- **Cypher Query:** `MATCH (f:Fund {name: "Aristotle Capital Management, LLC"})-[r:OWNS]->(c:Company) RETURN c.name, r.pct LIMIT 10`
- **Ground Truth:** American Water Works Company, I, 0.0323; Williams-Sonoma, Inc., 0.0765; Coterra Energy Inc., 0.040799998

## A.2. Multi-hop (2) Queries

These queries require traversing two relationships to find indirect connections, such as companies sharing the same institutional investor.

- **Question:** Which company is held by the same investment fund that owns Baxter International Inc. (ticker 'BAX')?
- **Cypher Query:** `MATCH (start:Company {ticker: 'BAX'})-[:OWNS]-(fund:Fund)-[:OWNS]-(target:Company) WHERE target.ticker <> 'BAX' RETURN target.name, target.ticker LIMIT 10`
- **Ground Truth:** Zimmer Biomet Holdings, Inc., ZBH; TE Connectivity plc, TEL; SBA Communications Corporation, SBAC; Regeneron Pharmaceuticals, Inc., REGN; Occidental Petroleum Corporatio, OXY

- **Question:** Identify the investment fund that holds a stake in Amgen Inc. (ticker 'AMGN') and name another company that this fund also owns.
- **Cypher Query:** `MATCH (start:Company)-[:OWNS]-(fund:Fund)-[:OWNS]-(other:Company) WHERE start.ticker = 'AMGN' RETURN fund.name AS fund_name, other.name AS other_company_name, other.ticker AS other_company_ticker LIMIT 10`
- **Ground Truth:** Capital International Investors, Broadcom Inc., AVGO; Capital International Investors, Builders FirstSource, Inc., BLDR

## A.3. Multi-hop (3) Queries

These queries require logical traversal of extended relationship chains, typically connecting disparate corporate entities through shared investors and personnel.

- **Question:** Trace the ownership dependencies originating from CME Group Inc. to identify the executive who works for the company at the end of the chain.

- **Cypher Query:** `MATCH (a:Company)-[:OWNS]-(b:Fund)-[:OWNS]-(c:Company)-[:WORKS_FOR]-(d:P` `WHERE a.ticker = 'CME' AND d.name IS NOT NULL RETURN d.name AS answer` `LIMIT 10`
- **Ground Truth:** Mr. David S. Marberger CPA; Ms. Tracy Schaefer; Ms. Noelle O'Mara; Ms. Melissa Napier; Ms. Charisse Brock

## A.4. Aggregation Queries

These queries test the system's numerical reasoning capabilities by applying global functions (`count`, `min/max`) to the extracted subgraph paths.

- **Question:** Compare the total number of news articles published about Ameriprise Financial, Inc. and Fortinet, Inc., returning each company's name and its respective article count ordered from highest to lowest.
- **Cypher Query:** `MATCH  (n:News)-[:NEWS_ABOUT_COMPANY]->(c:Company)  WHERE` `c.name IN ['Ameriprise Financial, Inc.', 'Fortinet, Inc.'] RETURN c.name` `AS company_name, count(n) AS article_count ORDER BY article_count DESC` `LIMIT 10`
- **Ground Truth:** Ameriprise Financial, Inc., 20; Fortinet, Inc., 14

- **Question:** Rank the top 3 funds by their ownership percentage in Ameriprise Financial, Inc. (AMP), returning the fund name and the exact percentage held.
- **Cypher Query:** `MATCH (f:Fund)-[o:OWNS]->(c:Company {ticker: "AMP"}) RETURN` `f.name AS fund_name, o.pct AS ownership_pct ORDER BY o.pct DESC LIMIT 3`
- **Ground Truth:** Vanguard Group Inc, 0.1331; JPMORGAN CHASE & CO, 0.1034; Blackrock Inc., 0.0968