# OpenReview forum: "Towards Ontology-Driven Multi-Hop Benchmark for Corporate GraphRAG Systems"
_ijcai.org/IJCAI-ECAI/2026/Workshop/GENAIK-NORA — IJCAI-ECAI 2026 Joint Workshop on GENAIK and NORA_

### Official Review · Reviewer_m1gQ · 2026-05-27
**Promising work but missing fine-grained details**

**Rating:** 5
**Confidence:** 3

**Review:**

Overall the work is interesting and shows a promising approach. There are, however, clarifications to be addressed to make it stand out.

# 3. Proposed Benchmark
- Figure 1 text is a bit blurry
- Figure 1: it is unclear what "YF" refers to. Acronyms should be explained for easier readability.
- Figure 1 needs explaining: it is unclear for example if "Wikipedia + GICS ontology" means that the Wikipedia ontology is given as well?

# 3.1.1 Formation of the Ontological Core
- What is the motivation/reason for using Qwen-7B as the LLM and not another model?

# 3.1.2 Dynamic Event Injection
- "First, static entities receive up-to-date event context that expands their profiles.": what is a profile here?
- "Second, news nodes begin to function as “dynamic bridges,” organically connecting different nodes through shared information events.": how? manually/through new relationships?

# 3.2.1 System Overviwe
- Figure 3 text is blurry

# 3.2.2 Schema Extraction and Data Grounding
- "Balanced sampling" seems very interesting but unclear how it is performed (e.g. python script?)
- "Because the theoretical existence of a path in a schema does not always guarantee its presence in concrete data,": a concrete example here would help better understand this point

# 4.2 Benchmark Quality Assessment
- Table 1: what is the explanation for the Simple case where LightRAG is the best on all metrics except Answer Relevance? This seems to go against the intuition of the authors when they proposed the metrics.

# Reproducibility
- GitHub link is included but the repository is empty

---

### Official Review · Reviewer_md8V · 2026-06-05
**GraphRAG and Corporate Multi-Hop QA**

**Rating:** 7
**Confidence:** 5

**Review:**

This is a very useful paper that provides some very useful results for practitioners in the field.  It has a few problems which could be easily addressed.  One is that describing what is meant by "mutli-hop questions" is not described, which is crucial.  A couple examples would be helpful.   Secondly, a description of GraphRAG itself is missing.

Examples of Aggregation and Analytical queries would also be helpful.

Secondly, if you are building a graph and an ontology, why is RAG needed at all?  How much of the questions addressed could be addressed with graph queries (SPARQL) alone?

A comparison against SOTA LLMs would also be helpful.

Are company subsidiary relationships represented?
You use "physical" when you mean "concrete" or "actual": physical solution -> concrete solution.

---

### Official Review · Reviewer_Zx7L · 2026-06-06
**Review for "Towards Ontology-Driven Multi-Hop Benchmark for Corporate GraphRAG Systems"**

**Rating:** 6
**Confidence:** 3

**Review:**

1. The abstract begins by identifying two evaluation problems; however, the main text appears to describe them more as objective phenomena rather than as explicit issues. It may be worth reconsidering the wording used to frame these key problems.

2. Do both GraphRAG and classical RAG face the static knowledge challenge mentioned in the paper?

3. Regarding the sentence “The use of such datasets is associated with a high risk of data contamination or memorization: the tested LLMs may retrieve correct answers from their own training data rather than from the provided RAG context [3]”, would it be useful to explain specifically why, under static data conditions, LLMs might directly produce correct answers from their training data instead of retrieving from the given RAG context?

4. In constructing the knowledge graph, the paper uses the temporal knowledge base dataset GDELT. Could extrapolation-based temporal knowledge graph completion methods also be integrated into the proposed approach?

---

### Official Review · Reviewer_xM5k · 2026-06-09
**Knowledge graphs expose limitations of contemporary RAG approaches**

**Rating:** 5
**Confidence:** 4

**Review:**

Constructing a knowledge graph from a static ontology with dynamic refreshes of some content within is not a novel approach by itself. Likewise, using knowledge graphs in the context of RAG is also not novel by itself. However, using paths and path traversals in knowledge graphs as the grounding prerequisite of QA benchmark generation is creative.

The paper tends to repeat that the execution of Cypher queries grounds QA pairs more often than is necessary, and that diminishes the effect. There are no examples in the paper, associating legitimate Cypher queries with paths in a knowledge graph and how such paths and subgraphs can answer either a few different questions or the same question asked in a few different ways. Evaluation uses expected metrics for retrieval and generation quality but the use of the same Qwen model for answer generation across different RAG approaches being evaluated seems a tad bit inadequate; quite likely that a different model might fare much better in this evaluation. The paper tries to justify use of the Qwen model but does not make further mention of other models that might be suitable alternatives, as 'future work'. The paper states often that the approach guarantees avoidance of LLM hallucinations in answer generation but it also uses LLMs to extract entity information based on strict ontology as a preparatory step in benchmark generation, so LLMs are not just used for answer generation (thus the guarantee about hallucinations, needs to be a bit more carefully qualified in writing).

Notably, there are no mentions of runtime performance, inference costs, analysis of errors observed and potential next steps to further this approach (e.g. customization, answer model evaluation,

Overall, a good attempt to enforce an improved fact-based answer generation, but there are multiple areas that can strengthen this publication a lot more, and I hope the authors take my feedback into consideration to come back with a much stronger paper.

---

### Decision · Program_Chairs · 2026-06-10

Accept